# Single-shot stereo-polarimetric compressed ultrafast photography for light-speed observation of high-dimensional optical transients with picosecond resolution

Jinyang Liang[1,2,3], Peng Wang[1,3], Liren Zhu[1] & Lihong V. Wang [ID] [1✉]

Simultaneous and efficient ultrafast recording of multiple photon tags contributes to high-dimensional optical imaging and characterization in numerous fields. Existing high-dimensional optical imaging techniques that record space and polarization cannot detect the photon's time of arrival owing to the limited speeds of the state-of-the-art electronic sensors. Here, we overcome this long-standing limitation by implementing stereo-polarimetric compressed ultrafast photography (SP-CUP) to record light-speed high-dimensional events in a single exposure. Synergizing compressed sensing and streak imaging with stereoscopy and polarimetry, SP-CUP enables video-recording of five photon tags ($x$, $y$, $z$: space; $t$: time of arrival; and $\psi$: angle of linear polarization) at 100 billion frames per second with a picosecond temporal resolution. We applied SP-CUP to the spatiotemporal characterization of linear polarization dynamics in early-stage plasma emission from laser-induced breakdown. This system also allowed three-dimensional ultrafast imaging of the linear polarization properties of a single ultrashort laser pulse propagating in a scattering medium.

[1] Caltech Optical Imaging Laboratory, Andrew and Peggy Cherng Department of Medical Engineering, Department of Electrical Engineering, California Institute of Technology, 1200 East California Boulevard, Mail Code 138-78, Pasadena, CA 91125, USA. [2] Present address: Laboratory of Applied Computational Imaging, Centre Énergie Matériaux Télécommunications, Institut National de la Recherche Scientifique, 1650 boulevard Lionel-Boulet, Varennes, QC J3X1S2, Canada. [3] These authors contributed equally: Jinyang Liang, Peng Wang. ✉email: LVW@caltech.edu

High-dimensional optical imaging is indispensable to maximally extract information carried by different photon tags[1]. These high-dimensional optical data are ubiquitously used in numerous fields of study, including biomedicine[2], agriculture[3], and electronics[4]. So far, most high-dimensional optical imaging systems acquire data through scanning. In brief, each measurement captures either a one-dimensional (1D) column or a two-dimensional (2D) slice[5,6]. By repeating measurements with varied parameters in the other dimensions, a high-dimensional array can be assembled. However, this scheme is subject to the assumption that the event is precisely repeatable[7] and also inherently suffers from a low measurement efficiency[8]. To overcome these barriers, many single-shot high-dimensional optical imaging techniques, in which multiple photon tags are measured in one acquisition, have been developed[9]. Such a parallel acquisition has maximized the captured information and measurement efficiency. Single-shot high-dimensional optical imaging modalities have experienced rapid growth in the past decade. Existing approaches can measure different combinations of photon tags besides 2D spatial information to enable three-dimensional (3D, e.g., spectral[10], volumetric[11], and temporal[12]), four-dimensional (4D, e.g., spectro-volumetric[13], plenoptic[14], and polarimetric[15,16]), and even five-dimensional (5D, e.g., spectro-polarimetric[17]) imaging.

As a subset of single-shot high-dimensional optical imaging, single-shot temporal imaging is of tremendous interest to many scientific communities. The ability to capture photon's time of arrival without repeating the measurement opens new routes for understanding underlying mechanisms in physics[18], chemistry[19], and biology[20] that are manifested in non-repeatable or difficult-to-reproduce events. However, the speed of light imposes imaging frame rates at the billion-frame-per-second (Gfps) level[21], far beyond the readout speed of the state-of-the-art charge coupled device (CCD) and complementary metal oxide semiconductor (CMOS) sensors[22]. To achieve a sub-nanosecond frame interval, various active illumination-based methods have been developed, including the frequency-dividing imaging[23], the angle-dividing imaging[24], sequentially-timed all-optical mapping photography[25], and frequency-domain streak imaging[26]. However, these methods have a low sequence depth (i.e., the number of captured frames in each acquisition). In addition, relying on engineered illumination schemes to provide ultra-high temporal resolutions, these methods are inapplicable to imaging a variety of luminescent and color-selective objects—such as distant stars, bioluminescent molecules, and scattering targets. On the other hand, single-shot passive ultrafast imaging has been achieved by utilizing a streak camera—a traditional 1D ultrafast imager that converts time to space by deflecting photoelectrons with a sweeping voltage perpendicular to the device's narrow entrance slit. Recent advances have enabled 2D ultrafast imaging by a pinhole array[27], a dimension reduction fiber bundle[28], and a tilted lenslet array[29]. However, these methods suffer from limited sampling in the field of view (FOV).

To surmount these problems, we develop an ultrafast single-shot high-dimensional imaging modality, termed stereo-polarimetric compressed ultrafast photography (SP-CUP). Synergistically combining compressed sensing, streak imaging, stereoscopy, and polarimetry, SP-CUP provides single-shot passive ultrafast imaging that can capture non-repeatable 5D [$x$, $y$, $z$: space; $t$: time of arrival; and $\psi$: angle of linear polarization (AoLP)], evolving phenomena at picosecond temporal resolution. Disruptively advancing existing CUP techniques[30–34] in imaging capability, SP-CUP enables simultaneous and efficient ultrafast recording of polarization in three-dimensional space. Compared with available single-shot ultrafast imaging techniques[35,36], SP-CUP has prominent advantages in light throughput, sequence depth, as well as spatiotemporal resolution and scalability in high-dimensional imaging.

## Results

**System and principle of SP-CUP.** The SP-CUP system (Fig. 1) consists of front optics, a dual-channel-generation stage, a spatial encoding stage, and two cameras (Equipment details are listed in Methods). The dynamic scene is first imaged by the front optics to the input image plane that interfaces the dual-channel-generation stage. The front optics varies the magnification according to the desired FOV in the specific study. In the dual-channel-generation stage, the incident light is first collected by a ×1 stereoscope objective lens, followed by a pair of diaphragms that sample the pupil to generate two optical channels (illustrated by the red and blue colors in Fig. 1a). Then, the light is sent through a pair of dove prisms that are rotated 90° with respect to one another. Following the dove prisms, a pair of beam splitters divide the light into two components (Fig. 1b). The reflected beam at each channel passes through a 50-mm-focal-length lens (L1a and L1b in Fig. 1b), are folded by a mirror and a knife-edge right-angle prism mirror (KRPM), and forms two images that are rotated 180° with respect to one another. An external CCD camera captures these two images as the two time-unsheared views (Views 1–2). The transmitted component, passing through the same configuration of lenses (L2a and L2b in Fig. 1b), mirrors, and KRPM, forms two images replicating those of the reflected beam. Then, these images are recorded by a compressed-sensing paradigm. Specifically, the light first passes through the spatial encoding stage. There, a 4$f$ imaging system, consisting of a 100-mm-focal-length tube lens (L3 in Fig. 1a) and a ×2 stereoscope objective, relays the images to a digital micromirror device (DMD). To spatially encode these images, a single pseudo-random binary pattern is displayed on the DMD. Each encoding pixel is turned to either +12° ("ON") or –12° ("OFF") from the DMD's surface normal and reflects the incident light to one of the two directions. The four reflected light beams, masked with complementary patterns, are collected by the same ×2 stereoscope objective. The collected beams are sent through two 75-mm-focal-length tube lenses (L4a and L4b in Fig. 1a), and folded by a pair of planar mirrors and a KRPM to form four horizontally aligned images in the streak camera. Inside the streak camera, the spatially encoded images successively experience temporal shearing by a sweep voltage applied to a pair of electrodes and spatiotemporal integration by an internal CMOS camera (detailed in Supplementary Note 1 and Supplementary Figure 1). The streak camera, therefore, acquires four time-sheared views (Views 3–6). Polarizers can be attached before the external CCD camera and at the entrance port of the streak camera to realize polarization sensing. Altogether, SP-CUP records six raw views of the scene in a single acquisition (detailed in Methods, Supplementary Note 2, and Supplementary Figure 2).

The image reconstruction of SP-CUP aims to recover the 5D information from the acquired six views in three steps. The first step reconstructs 3D ($x$, $y$, $t$) datacubes by combing selected views (see Supplementary Notes 3–4 and Supplementary Figure 3). Because a high-dimensional scene is recorded by 2D snapshots in data acquisition, direct inversion of the forward model is ill-conditioned. Thus, compressed sensing is implemented in the inverse problem, which solves Eq. 2 in Methods. In practice, a piece of reconstruction software that is developed from a compressed-sensing-based algorithm, termed the two-step iterative shrinkage/thresholding (TwIST) algorithm[37], is used to reconstruct the dynamic scene (detailed in Supplementary Note 3). Besides the high reliability of the implemented algorithms, two additional features assure the reconstructed image quality. First, the dove prism pair enables temporal shearing at two opposite angles. Along with the measurement from the external CCD camera, the high-dimensional scene is observed at three angles (Supplementary Figure 2). In addition, a total of six snapshots are recorded simultaneously, which remarks the highest data recording capability in all existing CUP

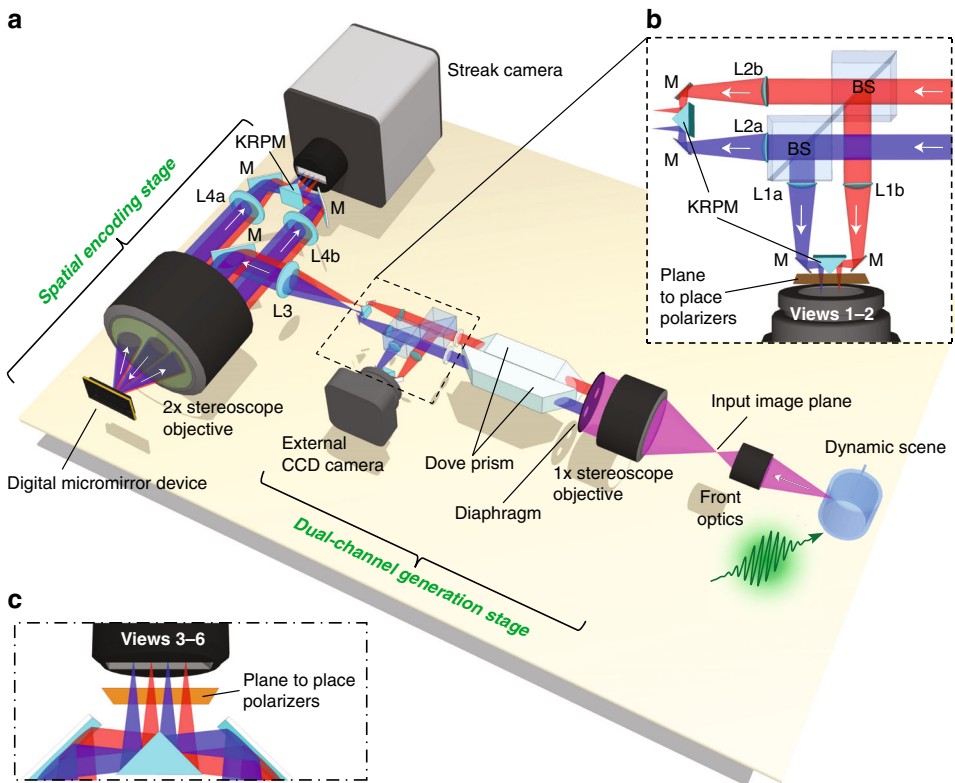

**Fig. 1 Schematic of the SP-CUP system.** BS, beamsplitter; L1a–L4b, Lenses; M, mirror; KRPM, knife-edge right-angle prism mirror. **a** The entire SP-CUP system. **b** Detailed illustration of beam splitting and polarization filtering in dual-channel-generation stage (dashed box). **c** Detailed illustration near the streak camera's entrance port. Equipment details are listed in Methods.

systems. Thus, the enriched observation to the transient scene and the increased data in measurement assist the reconstruction algorithm to stably and accurately recover the (x, y, t) datacubes from highly compressed measurement[38]. The details in system settings and characterizations are presented in Supplementary Notes 4 and 5.

The second step is to calculate the linear polarization parameters. Inserting polarization components in imaging systems is a commonly used method to passively detect polarization states of light. In general, based on the arrangement of linear polarizers specified by experiments, the recovered (x, y, t) information allows computing the spatiotemporal maps of the first three Stokes parameters (denoted by $S_0$, $S_1$, and $S_2$). For incident light composed of only a linearly polarized component and an unpolarized component, the time-resolved spatial distributions of the AoLP and the degree of linear polarization (DoLP) can be determined (explained in Supplementary Note 6).

As the last step, the recovered (x, y, t) datacubes are used to recover the depth information. The scheme of common-main-objective stereoscopy[39] is implemented in the dual-channel generation stage (Fig. 1a) to enable depth sensing (see Methods). This method does not rely on active illumination. Rather, it achieves passive detection without adding additional components. It is, therefore, selected to maximally leverage the existing design of the SP-CUP system. The spatial resolution, temporal resolution, and accuracy of polarization measurements are quantified under each setting used in the following experiments (detailed in Supplementary Notes 4–6).

**Plano-polarimetric ultrafast (x, y, t, ψ) imaging.** To detect four photon tags (i.e., x, y, t, ψ), we used SP-CUP to image a dynamic scene: five linear polarizers with different transmission angles were cut into the shapes of printed letters—"L", "S", "T", "U", and

"W"—and were overlaid on top of the corresponding prints. A 7-ps, 532-nm laser pulse illuminated these letters obliquely. The pulse was depolarized by a diffuser. Three 0º polarizers were inserted in Views 1, 3, and 4; three 45º polarizers were inserted in Views 2, 5, and 6. Combining Views 3 and 4 with View 1, we can use two projection angles to sense the transient scene filtered by the 0º polarizers. Similarly, the combination of Views 5 and 6 with View 2 enables reconstructing the same dynamics through the 45º polarizers. If light from the object is linearly polarized, the AoLP can be derived based on the intensity ratio between the recovered datacubes[40]. Meanwhile, the first Stokes parameter $S_0$, which represents the light intensity reflected by the object, can be readily obtained[41] (detailed in Supplementary Note 6).

Figure 2a shows the reconstructed ultrafast left-to-right sweeping dynamics at 250 Gfps. Four frames of the reconstructed ψ and $S_0$ are plotted in Fig. 2b–c, and the corresponding movies are presented in Supplementary Movie 1. The laser pulse swept through the sample at a measured apparent speed of $7 \times 10^8$ m s$^{-1}$, closely matching the theoretically expected value based on the pre-set experimental condition. The complete data (i.e., (x, y, t, ψ) information) is also visualized using the point cloud in Fig. 2d, showing that the five letters have distinct AoLPs, ranging across the π angular space. Figure 2e plots mean AoLPs ($\bar{\psi}$) over each letter versus time. The values of $\bar{\psi}$, averaged over time, for "L", "S", "T", "U", and "W" are 86.9º, –4.6º, 55.7º, –84.2º, and –44.9º, respectively. They are close to the measured ground truths: 84.8º, –1.8º, 53.9º, –83.7º, and –49.5º (black dashed lines in Fig. 2e). The standard deviation, averaged over time, ranges between 1.0º ("L") and 7.3º ("S"). Additional data are presented in Supplementary Figures 5 and 6.

**Characterization of early-stage plasma emission.** To demonstrate the indispensable utility of the SP-CUP system in ultrafast

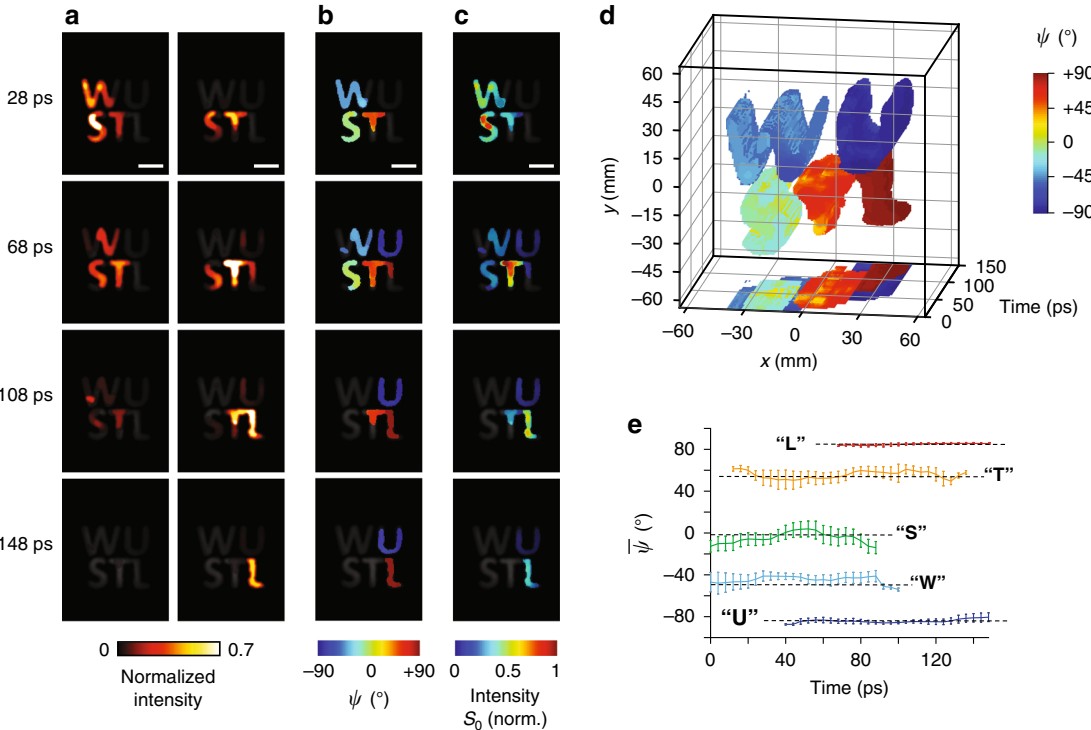

**Fig. 2 Plano-polarimetric ultrafast (*x, y, z, t, ψ*) imaging using SP-CUP. a** Representative frames of the reconstructed ultrafast dynamics at 250 Gfps from Views 1, 3, and 4 with the 0° polarizers (left column) and Views 2, 5, and 6 with the 45° polarizers (right column). The full sequence is in Supplementary Movie 1. These frames are overlaid on top of the time-unsheared image (grayscale) captured by the external CCD camera (Views 1 and 2). Note that the colormap is saturated at 0.7 in order to better display weak intensities. **b, c** Representative frames of the reconstructed AoLP (*ψ*) **b** and the first Stokes parameter ($S_0$) **c. d** Evolutions of *ψ* in the *x–y* plane. The projection on the *x–t* plane is plotted at the bottom. **e** Averaged values of *ψ* for the five letters over time. The black dashed lines are the ground truths measured using the time-unsheared images (Views 1 and 2). Note that the finite lengths of these dashed lines are for illustration only and do not indicate durations of events. Error bars in **e**, standard deviations. Scale bars in **a–c**, 30 mm.

physics, we used it to monitor the polarization dynamics of early-stage plasma emission in laser-induced breakdown (LIB). In our experimental setup (Fig. 3a), a single 800-nm femtosecond pulse (60-µJ pulse energy, 80-fs pulse width, 0° AoLP) was focused by a 20-mm-focal-length lens onto a silicon wafer [p-type (111)] in normal incidence. The silicon wafer was placed slightly before the lens focus to avoid the generation of plasma in air. The beam spot had a diameter of 20 µm at the surface of the silicon wafer, corresponding to a peak power density of 239 TW cm⁻². 

The plasma emission was collected by a ×10 objective lens, and the magnified image is formed on the input image plane. 0° polarizers were used for Views 1 and 3, and 45° polarizers were used for Views 2 and 5. Two-view reconstructions were carried out to recover the transient scene that is unfiltered, filtered by the 0° polarizers, and filtered by the 45° polarizers. Then, both AoLP and DoLP can be derived from the recovered datacubes, as described in Supplementary Note 6.

Shown in Fig. 3b–c and Supplementary Movie 2 are the plasma's polarization dynamics captured by SP-CUP in a single acquisition at an imaging speed of 100 Gfps and a sequence depth of 300 frames. The 4D imaging capability allowed us to analyze the angle-dependent expansion speed of the plume and the evolution of the polarization states of the plasma emission. SP-CUP captured the outward propagation of the plume front (yellow dashed lines) over time. The plume's expansion speed is nearly isotropic and the measured angle-averaged speed is 30 µm ns⁻¹ (Fig. 3d). Spatial evolutions of both AoLP and DoLP over six representative frames are also summarized in Fig. 3c. The spatially averaged DoLP is plotted in Fig. 3e. Both Figs. 3c, e suggest that DoLP of the plasma emission decreased from 0.58 at

time zero to about 0.33 at 3 ns. In addition, the AoLP, with a mean value of 0°, remains relatively unchanged in both spatial and temporal domains, as shown in Fig. 3c, f. More results are in Supplementary Note 8 and Supplementary Figures 7–9[40,41].

It is worth noting that this phenomenon is difficult to reproduce quantitatively[25]. As the event is non-repeatable, single-shot SP-CUP becomes especially advantageous in comparison to multi-shot pump-probe imaging methods. To illustrate this point, we compared the results of these two approaches (Supplementary Movie 3 and Supplementary Figure 9). In linearly fitting the plume's front position versus time, the pump-probe method shows a ~15× worse goodness of fit than SP-CUP. The substantial shot-to-shot variations of the plasma plume profile degrade the accuracy in plume's expansion speed calculation by 8.2×.

**Stereo-polarimetric ultrafast (*x, y, t, ψ*) imaging**. To detect five photon tags (i.e., (*x, y, t, ψ*)), we used SP-CUP to image the following dynamic scene at 100 Gfps: Three shapes—a square, a triangle, and a circle—were placed at three different depths. Similar to the plano-polarimetry ultrafast imaging experiment, each shape was covered by a linear polarizer of a distinct transmission angle, and the same laser pulse swept across these shapes at an oblique angle. 0° linear polarizers were inserted in Views 1, 3, and 4; 45° linear polarizers were inserted in Views 2, 5, and 6. Figure 4a shows nine representative snapshots of three-view reconstructions for both 0°- and 45°-filtered images (see the full sequence in Supplementary Movie 4). The apparent speed of laser pulse sweeping through the circle shape is measured $6 \times 10^8$ m s⁻¹, which is close to the prediction based on the experimental conditions. The result

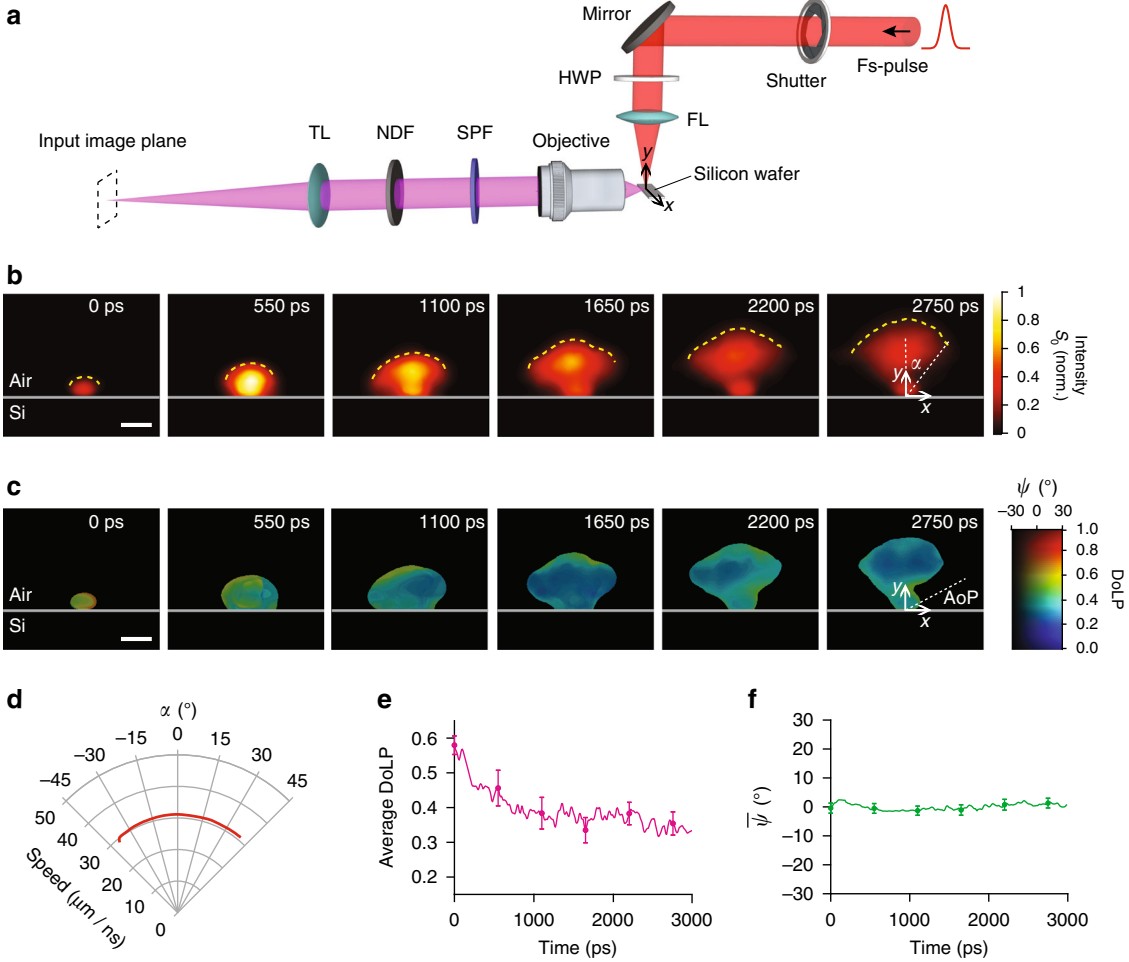

**Fig. 3 Spatiotemporally resolved polarization imaging of the plasma dynamics from the laser-induced breakdown on a silicon wafer. a** Experimental setup. The incident pulse was polarized in the $x$ direction. FL, focusing lens; HWP, half-wave plate; NDF, neutral density filter; SPF, short-pass filter; TL, tube lens. Equipment details are in Methods. **b** Normalized intensity ($S_0$) dynamics of a plasma plume. The gray line in each panel represents the air-silicon interface. The yellow dashed line represents the plasma plume front. The full sequence is shown in Supplementary Movie 2. **c** Spatiotemporally resolved DoLP and AoLP ($\psi$). In the 2D colormap, the jet and grayscale colormaps are used for DoLP and AoLP, respectively. **d** Plasma plume expansion speed over angle $\alpha$, which is defined in **b**. **e** Spatially averaged DoLP over time. **f** Spatially averaged $\psi$ over time. Scale bars in **b** and **c**: 50 μm. Error bars in **e** and **f** represent standard deviations of DoLP and $\psi$ in the spatial domain at the six snapshots shown in **c**.

shows that the three shapes were irradiated by the laser pulse one after another. Note that the circle in the 0°-filtered image has a low intensity because its polarization angle is nearly orthogonal to that of the 0° linear polarizer. These image pairs in Fig. 4a enable the stereo vision to sense depth (see Methods, Supplementary Note 9, and Supplementary Figures 10–13 for details). The reconstructed $\psi$ and $S_0$ distributions in 3D space at three representative frames (i.e., 90 ps, 130 ps, and 220 ps) are shown in Fig. 4b–c, respectively. The full evolutions are in Supplementary Movie 4. In addition, the 4D subsets of the complete data, namely $(x, y, z, t)$, $(x, y, t, \psi)$, and $(x, y, t, S_0)$ matrices, are displayed in Supplementary Figure 14.

Figure 4d–e provide the $z$ and $\psi$ values averaged over each shape. Both are nearly constant, which is consistent with the fact that these shapes were flat and covered with single polarizers. The reconstructed relative depths are 67 mm for the square at 90 ps, 103 mm for the triangle at 130 ps, and 151 mm for the circle at 220 ps. They are close to the ground truths of 70 mm, 102 mm, and 148 mm. The standard deviation of the depth, averaged over all shapes, is <5.5 mm, which is equivalent to ±0.2 pixel disparity accuracy. The circle shape has a larger standard deviation, as the greater difference in intensity between the image pairs compromised the accuracy in disparity calculation. A similar analysis is

applied to $\psi$ in Fig. 4e. The reconstructed $\psi$ values are −29.1° for the square at 90 ps, −0.9° for the triangle at 130 ps, and 83.0° for the circle at 220 ps. These measurements are also close to the ground truths of −27.7°, 1.9°, and 79.4°. The average standard deviation is ~4.5°. In Fig. 4f, $\overline{S_0}$ is the $S_0$ averaged over each shape. Following the ultrafast sweeping behavior of the laser pulse, $\overline{S_0}$ firstly rose and then fell in each shape. Additional results can be found in Supplementary Note 10 and Supplementary Figure 15.

**Observation of a laser pulse traveling in a scattering medium.** To highlight SP-CUP's single-shot 5D imaging capability in scattering media, we imaged the propagation of a single ultrashort laser pulse in water vapor at 100 Gfps. The dynamic nature of this scattering medium makes the experiment quantitatively non-repeatable. In this experiment, a light pulse of 400-nm wavelength and 55-fs width was guided into scattering water vapor. The incident beam was linearly polarized. Two metallic planar mirrors, arbitrarily placed inside the scattering medium, reflected the beam.

In the SP-CUP system, linear polarizers with its transmission angle aligned with that of the incident light were attached to Views 1, 3, and 4. No polarizers were attached to the other Views.

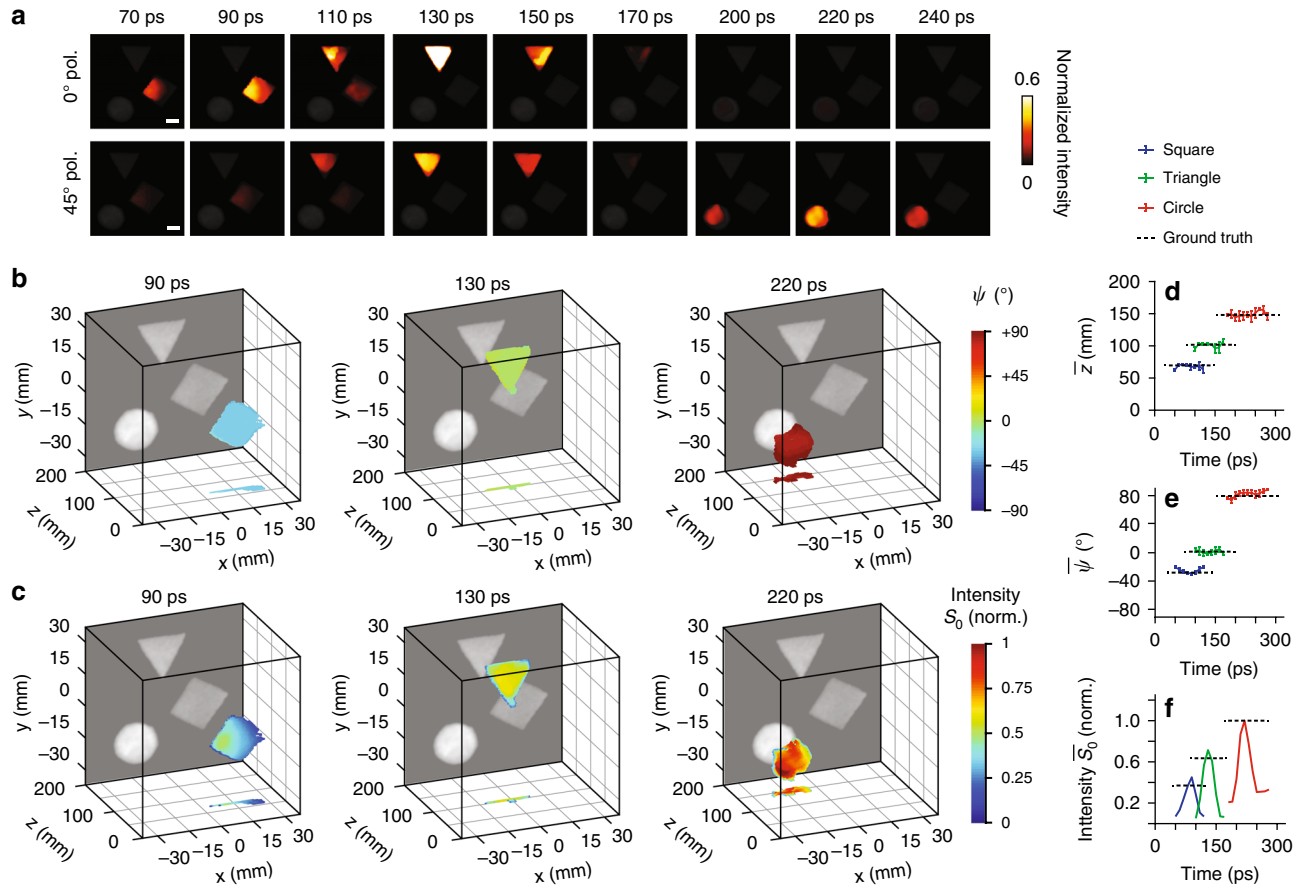

**Fig. 4 Stereo-polarimetric ultrafast ($x, y, z, t, \psi$) imaging using SP-CUP. a** Nine representative frames of the ultrafast dynamics imaged at 100 Gfps using Views 1, 3, and 4 (top row) through the 0° polarizers and Views 2, 5, and 6 (bottom row) thr°ugh the 45° polarizers. The full evolution is in Supplementary Movie 4. **b–c** Reconstructed AoLP ($\psi$) **b** and the first Stokes parameter ($S_0$) **c** in 3D space at three representative frames: 90 ps, 130 ps, and 220 ps. A relative depth is used by setting the 600-mm depth as the origin. The back side of the plot is the time-unsheared image. The images on the $x$–$z$ plane are the projections of the 3D plots along the $y$ axis. **d–f** Averaged values of $z$ **d**, $\psi$ **e** and $S_0$ **f** of the three shapes versus time. The black dashed lines are the ground truths measured using the time-unsheared stereo images from the external CCD cameras (Views 1 and 2). Note that the finite lengths of these dashed lines are for illustration only and do not indicate durations of events. Scale bar in **a**, 10 mm. Error bars in **d** and **e**, standard deviations.

Figure 5a shows the reconstructed light intensity in 3D space at seven representative frames (see the complete evolution in Supplementary Movie 5). Started by being short in time and narrow in space, the pulse becomes stretched gradually both temporally and spatially due to scattering from water vapor. The 3D trace of the centroids of the reconstructed pulse is in good agreement with the ground truth (Fig. 5b–c). The root-mean-square error of the centroid's position is close to 1% of the FOV on the $x$ and $y$ axes and 1% of the maximum depth on the $z$ axis (see Supplementary Figure 16 in Supplementary Note 11). We noticed the undesired scattering by the condensed water vapor on the mirror surface (e.g., the 700-ps frame in Fig. 5a), which obscured the location of the pulse centroid. Therefore, only those frames in which light propagated in straight paths were plotted in Fig. 5b–c.

For quantitative analysis, in Fig. 5d, the temporal profiles of light intensity at six selected spatial locations (labeled as $p_1$–$p_6$ in Fig. 5b) are given. The corresponding pulse durations are quantified in the inset. The temporal profile at $p_1$ had a full-width-at-half-maximum of 110 ps, as the pulse had already propagated some distance in the scattering medium before entering the FOV. The reconstructed pulse duration was gradually increased to 460 ps right before it exited the scattering medium, and it matched well with the direct streak camera measurement (Supplementary Figure 17). The beam cross-sections and the corresponding beam widths at five frames (labeled as $t_1$, $t_2$, $t_4$, $t_5$, and $t_7$ in Fig. 5a) are plotted in Fig. 5e.

The beam width was widened from 10 mm to 14 mm. The intensity of the pulse also decreased as expected. In addition, we found that scattering destroyed linear polarization (Fig. 5f). Owing to mirror surface scattering and the fact that mirror reflection alters the polarization state of light, we only studied the first 550 ps of light propagation. Both total intensity and DoLP decreased over time (dots), and single-component exponential fits (solid lines) yielded decay constants of 2.5 ns and 9.1 ns, respectively.

It is worth noting that the result presented here is achievable only by using SP-CUP, as demonstrated in Supplementary Figure 18. When the transient event occurs in 3D space, the previous state-of-the-art CUP systems[30–32], capable of only 2D ultrafast imaging, inevitably suffer from inaccuracy in the detection of the time of arrival. In contrast, with the powerful 5D imaging ability, SP-CUP has successfully recorded the full evolution of pulse propagation with clearly resolved 3D spatial positions and correctly detected times of arrival.

## Discussion

SP-CUP is able to improve the capabilities of imaging LIB dynamics. At present, most techniques (e.g., LIB spectroscopy) measure LIB dynamics by integrating signals over space and time and by averaging over multiple laser shots. However, these approaches confront measurement inaccuracy in many scenarios,

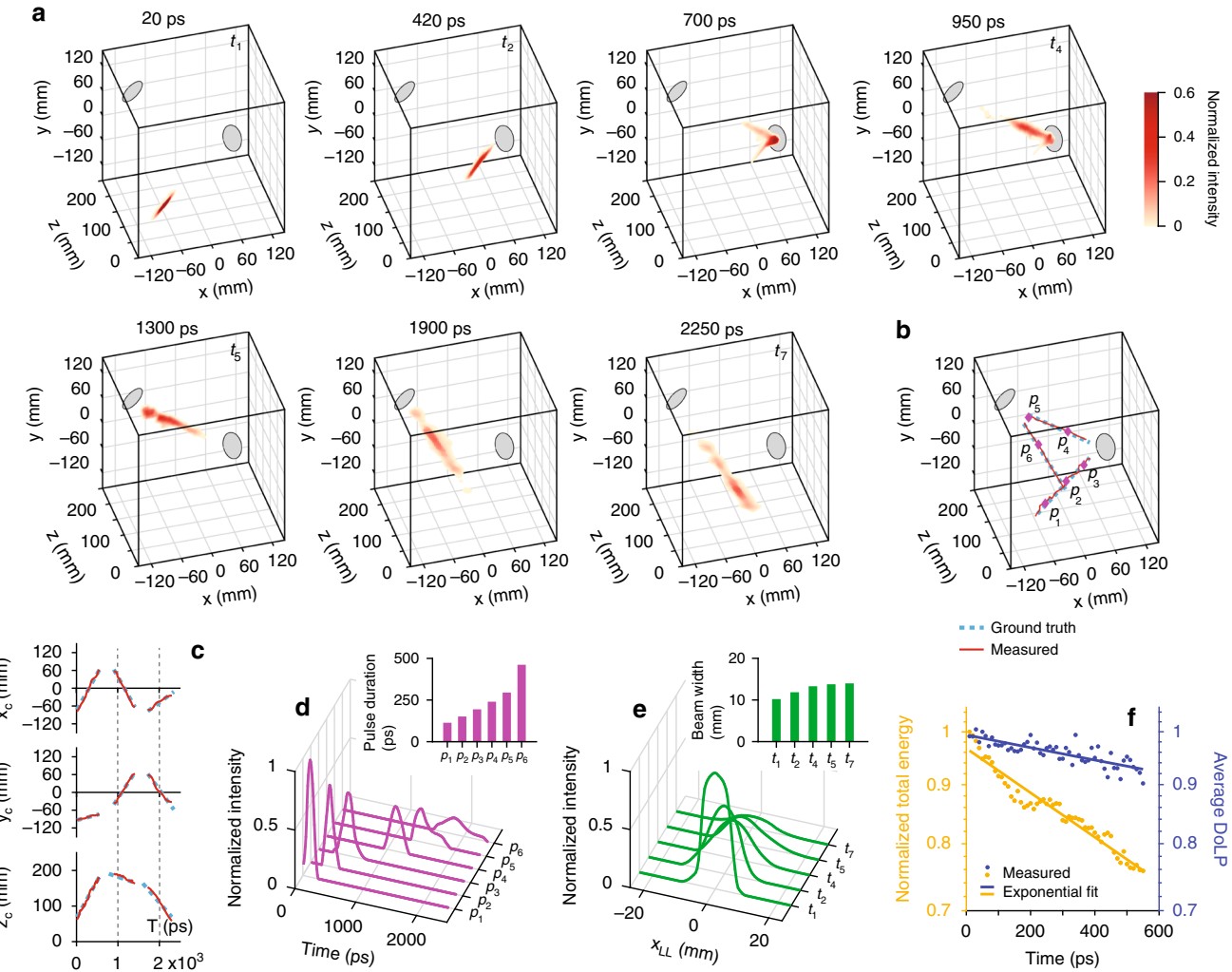

**Fig. 5 Stereo-polarimetric ultrafast imaging of an ultrashort laser pulse propagation in a 3D scattering medium. a** Seven representative frames of the ultrashort pulse in 3D space, imaged at 100 Gfps. The incident pulse was initially polarized along the $y$ direction. The full movie is in Supplementary Movie 5. A relative depth is used by setting the 600-mm depth as the origin. **b**, **c** Traces of both reconstructed (red solid line) and ground truth (cyan dashed line) centroids of the pulse in 3D space **b** and in three orthogonal spatial dimensions **c**. **d** Evolutions of light intensity at six spatial locations ($p_1$–$p_6$) labeled as diamonds in **b**. Inset: the measured pulse durations at these six locations. **e** Cross-sections of reconstructed beam intensity in local lateral dimension ($x_{LL}$) at five time points ($t_1$, $t_2$, $t_4$, $t_5$, and $t_7$) labeled in **a**. Inset: the measured beam widths at these five time points. **f** Reconstructed total intensity integrated over 3D space (left vertical axis and yellow dots) and reconstructed DoLP (right vertical axis and blue dots) before the pulse was incident to the first mirror. Single-component exponential fits were applied to both data sets (solid lines). They are plotted on the log scale.

including sample variations[42], changes of the physical and chemical properties of the sample surface[43], and laser-fluctuation-induced variations in plasma's characteristics[44]. To overcome these limitations, single-shot LIB imaging[24,44], time- and/or space-resolved LIB imaging[45,46], and polarization-resolved LIB imaging[47] have been developed individually. These developments suggest that leveraging all the information from space, time, and polarization in a single-shot acquisition appears to be an optimal LIB detection strategy[48]. However, to our knowledge, no existing technique could meet this requirement. SP-CUP fills this void by resolving transient light intensity and polarization evolutions in space (Fig. 3). Consequently, SP-CUP offers a new approach to improving the measurement accuracy over the multi-shot counterpart (see Supplementary Note 8 and Supplementary Movie 3).

Besides demonstrating the attractive merits of single-shot spatiotemporally resolved polarization imaging (Fig. 3), it is imperative to explore the elusive origin of polarized emission in LIB[49]. Various theoretical models have been established to deduce possible sources of the observed polarization[50,51]. Among them,

the theory based on an anisotropic electron distribution hypothesizes polarized emission on a picosecond time scale, followed by a decreased degree of polarization over time as plasma comes to a steady state. Previous research, utilizing systems of nanosecond temporal resolution[52], could not distinguish the plasma emission in such an early stage. In contrast, the SP-CUP system, with a picosecond-level temporal resolution, directly observed decreased DoLP of the plasma emission in the early stage (i.e., 0–3 ns). Furthermore, we observed that the AoLP of the emission remained constant with a small (±2.5°) fluctuation[53]. Our results suggest that the polarization in the plasma emission could be determined by the anisotropic electron distribution function in the plasma that decreases homogeneously over time[53,54].

We have demonstrated SP-CUP—an ultrafast single-shot high-dimensional imaging modality—with an imaging speed up to 250 Gfps, a sequence depth up to 300 frames, and a ($x$, $y$) pixel count of 0.3 megapixels (500 × 600 pixels) per frame. Moreover, by deploying different polarization components in different views, the AoLP and DoLP were accurately measured. Finally, the adoption of

stereoscopy enabled simultaneous extraction of five photon tags $(x, y, z, t, \psi)$ from the compressively acquired snapshots. Collectively, SP-CUP features high light throughput, high spatial and temporal resolutions, a large sequence depth, as well as its attractive ability to image non-repeatable and difficult-to-reproduce high-dimensional transient events. SP-CUP's specifications could be further upgraded by using an optimized mask[55] and a femtosecond streak tube with a large photocathode[30]. Moreover, by inserting other polarization components (e.g., a quarter-wave plate) into particular channels, the SP-CUP system could measure the full set of the Stokes parameters[56] that characterizes light in arbitrary polarization states. Finally, new algorithms based on deep neural networks[57] could improve the speed of high-dimensional reconstruction while retaining the reconstruction fidelity. SP-CUP could be applied to many areas of studies, including time-resolved LIB micro-spectroscopy for composition analysis in organic chemistry and biology[58,59] and sonochemistry in marine biology[60–62]. All of these directions are promising research topics in the future.

## Methods

**Equipment details of the SP-CUP system.** The components, in Fig. 1 in Main Text, include a beamsplitter (Thorlabs, BS013), a DMD (Texas Instruments, LightCrafter 3000), dove prisms (Thorlabs, PS994), an external CCD camera (Point Gray, GS3-U3-32S4M-C), an internal CMOS camera (Hamamatsu, Flash-4.0), L1a, L1b, L2a, and L2b (Thorlabs, AC254-050-A), L3 (Thorlabs, AC508-100-A), L4a and L4b (Thorlabs, AC254-075-A), knife-edge right-angle prism mirrors (Edmund Optics, 49-413), and a streak camera (Hamamatsu, C7700). The two apertures on the diaphragm have diameters of 2.54 cm to match the size of the dove prisms. The use of diaphragm can also prevent stray light from entering the system from the ×1 stereoscope objective lens.

The setup to generate plasma on the silicon wafer, in Fig. 3a in Main Text, includes a mechanical shutter (Vincent Associate, Uniblitz LS3Z2), a half-wave-plate (Thorlabs, WPH05M-780), a focusing lens (Thorlabs, LA1074), an objective (Thorlabs, RMS10X), a tube lens (Thorlabs, LBF254-200), a short-pass-filter (Thorlabs, FGB37M), and a neutral density filter (Thorlabs, NE20A).

The principle of the streak camera is illustrated in Supplementary Figure 1. The input optics first relays the image of the fully opened entrance slit (17 mm × 5 mm) to the photocathode on a streak tube. The photocathode converts the photons into photoelectrons. After gaining sufficient speed through an acceleration mesh, these photoelectrons experience a time-varying shearing operation by a sweep voltage on the vertical axis, according to their times of arrival. Then the temporally sheared photoelectrons are converted back into photons by using a phosphor screen. Finally, an image intensifier boosts the optical signal before it is captured by an internal CMOS camera in a single 2D image.

**Data acquisition of SP-CUP.** In data acquisition, a transient scene is imaged in two time-unsheared views and four time-sheared views (detailed in Fig. 1, Supplementary Note 2 and Supplementary Figure 2). The time-unsheared views directly record the scene akin to conventional photographs. The measured optical energy distributions are denoted as $E_1$ and $E_2$. Each of the four time-sheared views uses the compressed-sensing paradigm to record the scene. Specifically, in each time-sheared view, the scene is first spatially encoded, then temporally sheared, and finally spatiotemporally integrated. The measured optical energy distributions are denoted as $E_3$, $E_4$, $E_5$, and $E_6$, respectively. Mathematically, the six raw views can be linked to the intensity distribution of a dynamic scene $I(x, y, t)$ as follows:

$$\begin{bmatrix} E_1 \\ E_2 \\ E_3 \\ E_4 \\ E_5 \\ E_6 \end{bmatrix} = \begin{bmatrix} T\boldsymbol{F}_1 \\ \boldsymbol{R_d}T\boldsymbol{D}_2\boldsymbol{F}_2\boldsymbol{R_p} \\ TS\boldsymbol{D}_3\boldsymbol{F}_3\boldsymbol{C}_{[10]} \\ TS\boldsymbol{D}_4\boldsymbol{F}_4\boldsymbol{C}_{[01]} \\ \boldsymbol{R_d}TS\boldsymbol{D}_5\boldsymbol{F}_5\boldsymbol{C}_{[10]}\boldsymbol{R_p} \\ \boldsymbol{R_d}TS\boldsymbol{D}_6\boldsymbol{F}_6\boldsymbol{C}_{[01]}\boldsymbol{R_p} \end{bmatrix} I(x, y, t) \qquad (1)$$

where $\boldsymbol{T}$ represents spatiotemporal integration, $\boldsymbol{F}_j(j = 1, \ldots, 6)$ represents spatial low-pass filtering due to the optics of View $j$, $\boldsymbol{R_d}$ represents the 180° digital image rotations conducted by the external CCD camera and the streak camera, $\boldsymbol{D}_i(i = 2, \ldots, 6)$ represents image distortion with respect to View 1, $\boldsymbol{R_p}$ represents the 180° physical image rotation induced by the dove prism, $\boldsymbol{S}$ represents temporal shearing, and $\boldsymbol{C}_{[10]}$ and $\boldsymbol{C}_{[01]}$ represent the spatial encoding of a pair of complementary masks realized by the "ON" and "OFF" DMD pixels, respectively. Besides these general operations, stereoscopic sensing (denoted by operators $\boldsymbol{Z_L}$ and $\boldsymbol{Z_R}$) and the polarization sensing (denoted by operators $\boldsymbol{P_0}$ and $\boldsymbol{P_{45}}$) are incorporated in the imaging process by configurations chosen for specific experiments (detailed in Supplementary Note 4).

**Image reconstruction of SP-CUP.** Given the known operators and the spatio-temporal sparsity of the dynamic scene, we recover $I(x, y, t)$ by solving the following inverse problem

$$\hat{I} = \mathrm{argmin}_I \left\{ \frac{1}{2} \|E - \boldsymbol{O}I\|_2^2 + \beta \Phi_{\mathrm{TV}}(I) \right\}, \qquad (2)$$

where $E$ represents the joint measurement of energy distributions of selected views required by a specific experiment, and the joint operator $\boldsymbol{O}$ accounts for all operations from each selected view. (detailed in Supplementary Note 4). The first term $\frac{1}{2}\|E - \boldsymbol{O}I\|_2^2$, where $\|\cdot\|_2$ denotes the $l2$ norm, quantifies the difference between the measurement $E$ and the estimate using the reconstructed datacube $\boldsymbol{O}I$. $\Phi_{\mathrm{TV}}(I)$ is the total variation (TV) regularizer that promotes sparsity in the dynamic scene. The regularization parameter $\beta$ adjusts the weight ratio between the two terms to produce the reconstruction that best complies with the physical reality. A pre-processing algorithm developed in house (detailed in Supplementary Note 3) fully aligns the spatial positions of all views and calibrates the energy ratios between them. Then, $E$ is sent into a software based on the TwIST algorithm to reconstruct the dynamic scene[41].

SP-CUP provides projection views of the dynamic scene from three different angles (Supplementary Figure 2). First, the two time-unsheared views (i.e., Views 1–2) record only spatial information without temporal information. Therefore, the projection angle, determined by unsheared temporal integration, is parallel to the $t$ axis. Second, the first and second time-sheared views (i.e., Views 3–4) record both spatial and temporal information through temporal shearing. The spatiotemporal integration is operated on a spatially encoded and temporally sheared datacube. In the original datacube (see the matrix labeled with $\boldsymbol{I}$ in the center of Supplementary Figure 2), the aforementioned spatiotemporal integration operation is equivalent to integrating along a tilted direction that is oblique to the $t$ axis (see the matrix at the top right corner of Supplementary Figure 2). Third, following the same scheme, the third and fourth time-sheared views (i.e., Views 5–6) form the third projection view. However, because the dove prism induces a 180° image rotation, the integration direction is opposite to that in the second projection view (see the matrix at the bottom right corner of Supplementary Figure 2). Overall, SP-CUP enriches the observation by providing three distinct projection views of the dynamic scene.

**Operating principle of stereoscopy.** The left and right images from the dual-channel generation stage were captured and reconstructed by Views 2, 5, and 6 and Views 1, 3, and 4, respectively. We first calibrated the imaging system for the geometric parameters, primarily the distortions ($\boldsymbol{D}_2$ in Eq. 1 in Methods), focal length ($f$), and baseline separation between the left and right images ($l$) (detailed in Supplementary Note 9 and Supplementary Figures 10–13). Here, the baseline, which connects the centers of the stereoscopic images, is in the horizontal direction. Semi-global block matching was employed to find a map of $\delta$, which is the disparity between the left and right images (see Supplementary Note 9 for their definitions), in the unit of pixels. A larger $\delta$ means that the object is closer and vice versa. Subsequently, depth was computed via

$$z = \frac{fl}{\delta d}, \qquad (3)$$

where $d$ is the sensor's pixel size. Together with the polarimetric sensing ability, the SP-CUP system can record five-dimensional data containing $(x, y, z, t, \psi)$ information in a single shot.

## Data availability
The data that support the findings of this study are available from the corresponding author on reasonable request.

## Code availability
The reconstruction algorithm is described in detail in Methods and Supplementary Information. We have opted not to make the computer code publicly available because the code is proprietary and used for other projects.

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

## Acknowledgements

The authors thank Dr. Chiye Li and Dr. Pengfei Hai for experimental assistance, and Professor Liang Gao from the University of Illinois at Urbana-Champaign for fruitful discussion. This work was supported in part by National Institutes of Health grants DP1 EB016986 (NIH Director's Pioneer Award) and R01 CA186567 (NIH Director's Transformative Research Award).

## Author contributions

J.L. proposed the high-dimensional CUP imaging and designed the SP-CUP system. J.L. and P.W. built the system. P.W. and J.L. performed the experiments. P.W., L.Z., and J.L. developed the SP-CUP's image reconstruction algorithm. J.L., P.W., and L.Z. analyzed the experimental data and drafted the manuscript. L.V.W. conceived compressed-sensing-aided multi-view projection and supervised the project. All authors revised the manuscript.

## Competing interests

The authors disclose the following patent applications: WO2016085571 A3 (L.V.W. and J.L.), US Provisional 62/298,552 (L.V.W., J.L., and L.Z.), and US Provisional 62/813,974 (L.V.W., P.W., J.L., and L.Z.).
