## [Peer Review File · Nature Communications]

REVIEWERS' COMMENTS:

Reviewer #1 (Remarks to the Author):

The authors present highly innovative work on a novel imaging platform capable of recording both stereoscopic and polarization information with picosecond temporal resolution. The method is based on the streak camera approach, which converts temporal resolution into a spatially sheared image. Although conventional streak cameras produce a one-dimensional image over time, the authors use compressive sensing methodology to produce 2-D image by imaging spatially sparse scenes. This work further pushes the frontier of ultra-fast imaging by capturing polarization information while also computing depth of the imaged objects.

The authors demonstrated that with this new and exciting technology, new physical phenomena, which have never been observed before, can be examined. In particular, polarization effects related to early-stage plasma are observed and reported for the first time in the literature. These effects last for several hundreds of picoseconds and have never been studied before. These experiments demonstrated that this imaging technology will help uncover new basic science phenomena that have not been able to be observed before due to technology limitations. This work is highly innovative and transformative.

There are few minor comments that can help improve the quality of the manuscript:

1. The title of the paper suggests "real-time", which typically implies 30 to 100 frames per second temporal resolution. Since this technology is capturing events at over billion frames per second, the real-time does not accurately depict the temporal capabilities of the system. I suggest including either picosecond resolution or billion frame-rate instead.
2. Also the work focused on linear polarization while the title of the paper states "polarization". It would be good to add linear to discern that circular polarization is not currently captured with the presented system.
3. The conclusion of the paper states that the angle and degree of linear polarization were accurately measured though error metric (or accuracy and precision) for polarization imaging is not presented. The authors should quantify the error in the reconstructed angle and degree of linear polarization for different light intensities and present it in the paper.
4. Related to the previous point, in the measured angles of polarization in the various experiments, the authors state the difference between the measured and actual values. What is this error due to and does the system need to be calibrated with respect to polarization (similar to the calibration used for stereoscopic imaging)? Discussion on this error should be included in the paper and how it can be mitigated.
5. In figure 3-c, the color scheme does not capture very well the temporal variations in the DOLP because they are mostly represented as grayscale. Since the AOP does not change, it would be better to represent the DOLP in jet scale and AOP as grayscale. This can help in better visualization of the ~30% drop of DOLP.

Reviewer #2 (Remarks to the Author):

This is an interesting paper. I recommend publication with mandatory revisions, primarily aimed at clarifying the methods and the data.

The description of the image reconstruction given on pages 3-5 is not transparent enough for the general audience of Nature Communications. I recommend that the authors include a more intuitive description. Leaving the details of the methods to the Methods and Supplementary Materials is OK, but the reader should be able to have a fairly good understanding of how the method works based on the description in the main text. Perhaps the authors can start with an intuitive description of their basic CUP technique and then explain how they add the additional z and polarization dimensions?

Why is the inverse problem under-determined in this case, and how that affects the image retrieval?

The authors need to include the estimates for the limitations of their imaging technique, within the

implementation used here – the x,y,z,t resolutions and ranges, as well as the error of the measurement of the DoLP. What are the major limiting factors? That would give the reader an idea of what type of dynamic processes this technique can be used to image.

In my opinion, including the description of the operation of a streak camera in the Supplementary seems unnecessary. That is a common knowledge.

The author's example involving the polarization evolution of the emission by the ablation plume is interesting, if they are looking at a real effect. First, the ejecta they are showing are not "early-stage plasma," as stated. In fact, the free electrons in the plume are most likely recombined or attached themselves to atoms or oxygen molecules in air within few tens of ps of excitation. It is still not clear why this emission should be polarized. It is commonly believed that this emission (LIBS emission) results from the impact excitation of the ejected atoms and ions by free electrons, followed by unpolarized fluorescence from excited states of those atoms and molecules. The values for the degree of polarization that authors report are not very large. Are they within the measurement error of their method? In other words, are the authors looking at a real effect?

The relevant data shown in Figure 3(c) are not clear. Color coding in two dimensions (DoLP and psi) may not be a good representation of the data in this case. The plots look predominantly grey. Perhaps the authors can change the color mapping or present the data in some other way to make the features they are talking about stand out better?

If the measured DoLPs are within the measurement error, what could be a possible physical explanation for this effect? What defines the AoP and why is it changing (or not changing – it is not clear from the data) with time?

Reviewer #3 (Remarks to the Author):

Introduction and Novelty

The manuscript introduces stereo-polarimetric compressed ultrafast photography (SP-CUP), an imaging system capable of recording high-dimensional events in a single exposure at speeds surpassing other single-shot high-dimensional cameras in the literature. The readout speed of current state-of-the-art sensors has limited single-shot temporal imaging to speeds significantly lower than the speed of light. However, SP-CUP bypasses this issue by implementing streak imaging. The system, however, also avoids inherent problems with streak imaging such as low sequence depth and a low field of view as well as producing high dimensional data by combining it with elements of stereoscopy, and polarimetry, and compressed sensing. The manuscript first describes the optical setup used to achieve these results in detail before moving on to a series of experiments that demonstrate their model's ability to capture multiple photon tags and generalizability to different applications. This method is novel in its advancement of existing CUP techniques by enabling simultaneous recording of polarization in three-dimensional space. It also improves upon current methods of single-shot ultrafast imaging techniques by producing a more favorable sequence depth, spatiotemporal resolution, and scalability in high dimensional imaging. The proposed method is clearly described in the context of alternate options and its novel improvements are outlined in detail.

Major Critiques

This manuscript makes it clear that SP-CUP can enable the collection of multiple photon tags in scattering media, and the recording of non-repeatable, single-shot events such as the polarization dynamics of plasma in laser-induced breakdown. The value that this device brings to the scientific community is evident, however, several aspects of this manuscript should be modified for it to match the caliber of other manuscripts published in Nature Communications. Firstly the paper mentions several times that SP-CUP synergizes compressed sensing and streak imaging with stereoscopy and polarimetry. Not only does this statement become redundant throughout the manuscript, but its implications are not fully explored. How streak imaging, stereoscopy, polarimetry, compressed sensing are integrated into the larger system is made clear, however, why these techniques, in particular, were selected and how they contribute to the advancements

that the system has made should be fully fleshed out to fully orient the reader. Furthermore, the optical setup figure should be more integrated into the main body of text with perhaps less text description and more references to Figure 1 so that it's easier to visualize what the manuscript is referring to when describing each component.

The discussion section of the paper introduces more applications where SP-CUP could be implemented. This section is lengthy. Though applications discussed should be mentioned, without quantitative evidence for how SP-CUP performs in these environments, an in-depth write up lacks value to justify its length.

Overall, the manuscript in its current format is not yet suitable for publication without major modifications as suggested above. Should the authors consider addressing those comments, the value of SP-CUP can be made succinct and obvious to the scientific community, and I would gladly review again an updated version of the manuscript.

RESPONSES TO REVIEWERS

We sincerely appreciate the reviewers' timely effort in providing valuable comments of this manuscript, especially under the exceptional circumstance of COVID-19. We have adopted these comments to improve the quality of our manuscript. The point-by-point responses are listed as follows. All changes in the revised manuscript are highlighted in red.

REVIEWER 1

[Comment 0]

The authors present highly innovative work on a novel imaging platform capable of recording both stereoscopic and polarization information with picosecond temporal resolution. The method is based on the streak camera approach, which converts temporal resolution into a spatially sheared image. Although conventional streak cameras produce a one-dimensional image over time, the authors use compressive sensing methodology to produce 2-D image by imaging spatially sparse scenes. This work further pushes the frontier of ultra-fast imaging by capturing polarization information while also computing depth of the imaged objects.

The authors demonstrated that with this new and exciting technology, new physical phenomena, which have never been observed before, can be examined. In particular, polarization effects related to early-stage plasma are observed and reported for the first time in the literature. These effects last for several hundreds of picoseconds and have never been studied before. These experiments demonstrated that this imaging technology will help uncover new basic science phenomena that have not been able to be observed before due to technology limitations. This work is highly innovative and transformative.

[Response 0]

We sincerely appreciate the reviewer's acknowledgment of the technical innovation carried in our work. We also thank the review for her/his recognition of the application of SP-CUP to observing new physical phenomena.

There are few minor comments that can help improve the quality of the manuscript:

[Comment 1]

1. The title of the paper suggests "real-time", which typically implies 30 to 100 frames per second temporal resolution. Since this technology is capturing events at over billion frames per

second, the real-time does not accurately depicts the temporal capabilities of the system. I suggest including either picosecond resolution or billion frame-rate instead.

[Response 1]

In our manuscript, “real time” is defined, according to the Merriam-Webster dictionary, as “the actual time during which something takes place” [R1]. Therefore, the word is properly chosen for the title.

However, we agree with the reviewer that the word “real time” has different interpretations in the imaging community. Therefore, following the reviewer’s suggestion, we have changed the title to “Single-shot stereo-polarimetric compressed ultrafast photography: Light-speed observation of high-dimensional optical transients with picosecond resolution”. We also defined “real time” in the revised abstract.

[Comment 2]

2. Also the work focused on linear polarization while the title of the paper states “polarization”. It would be good to add linear to discern that circular polarization is not currently captured with the presented system.

[Response 2]

We have revised throughout the manuscript to clearly state that the polarization parameters that SP-CUP system captures are the angle of linear polarization (AoLP) and the degree of linear polarization (DoLP). The word “linear” has been added throughout the revised manuscript to emphasize the type of polarization that the system can measure. Finally, we have added a sentence in Summary about future work on extending SP-CUP to measure light with arbitrary polarization states (see Lines 323–325, Page 11, in the revised manuscript).

However, we decided to keep “stereo-polarimetric” in the title for generality and simplicity. The current system can capture both depth and polarization information in addition to 2D (x, y) space and time. The reviewer is correct that our current system can only detect linear polarization, however, with modifications (e.g., by including more views with more different polarizers), an upgraded system may enable detection of the polarization state of any light.

[Comment 3]

3. The conclusion of the paper states that the angle and degree of linear polarization were accurately measured though error metric (or accuracy and precision) for polarization imaging is not presented. The authors should quantify the error in the reconstructed angel and degree of linear polarization for different light intensities and present it in the paper.

[Response 3]

The related data have been presented in our original manuscript. The error quantification of measured AoLPs in SP-CUP was conducted with the two proof-of-concept experiments (see Sections 2 and 4 in Results). In particular, for plano-polarimetric ultrafast imaging, this information is presented in Lines 169–170 on Page 6 in the original manuscript. For stereo-polarimetric ultrafast imaging, this information is presented in Lines 235 on Page 8 in the original manuscript. Because we covered linear polarizers on top of these letters and shapes, the light incident into the SP-CUP system is linearly polarized.

Following the reviewer's suggestion, we have analyzed the DoLP accuracy under different signal-to-noise ratios (SNRs). This new analysis has demonstrated that SP-CUP can accurately quantify the DoLP. The results and analyses are now included in Lines 885–915 on Pages 41–43 in Supplementary Note 6.

[Comment 4]

4. Related to the previous point, in the measured angles of polarization in the various experiments, the authors state the difference between the measured and actual values. What is this error due to and does the system need to be calibrated with respect to polarization (similar to the calibration used for stereoscopic imaging)? Discussion on this error should be included in the paper and how it can be mitigated.

[Response 4]

The errors in the calculation of AoLP and DoLP are mainly induced by the noise performance of the streak camera and by the reconstruction algorithm. The streak camera induces noise during the photon-to-photoelectron conversion in the photocathode, photoelectron-to-photo conversion on the phosphor screen, and the optical signal amplification inside the image intensifier. All these random noise sources cannot be calibrated. On the other hand, strong illumination intensity would aggravate the space-charge effect, which blurs the streak image. As a result, the SNR in the raw data is moderate. The noise in the raw data affects how accurately the reconstruction algorithm

allocates the signals to the correct spatiotemporal voxel in the reconstructed movie. Finally, the intensity error in the reconstructed movie is transferred to the inaccuracy in the calculation of AoLP and DoLP.

The mitigation of the errors in AoLP and DoLP can be performed by tackling these noise sources. Possible approaches include modifying the streak tube's design, using optical-streaking approaches, limiting the photon flux, and implementing new penalty terms and/or regularizers. The related discussion has been included in Supplementary Note 6 (see Lines 918–935 on Page 43 in the revised manuscript).

[Comment 5]

5. In figure 3-c, the color scheme does not capture very well the temporal variations in the DOLP because they are mostly represented as grayscale. Since the AOP does not change, it would be better to represent the DOLP in jet scale and AOP as grayscale. This can help in better visualization of the ~30% drop of DOLP.

[Response 5]

Following the reviewer's suggestion, we have re-plotted Fig. 3c and Supplementary Movie 2 by using a jet colormap for DoLP and a grayscale colormap for AoLP.

[Reference]

[R1] "Real time." in *Merriam-Webster.com Dictionary*, Merriam-Webster, <https://www.merriam-webster.com/dictionary/real%20time>. Accessed 15 Jun. 2020.

REVIEWER 2

[Comment 0]

This is an interesting paper. I recommend publication with mandatory revisions, primarily aimed at clarifying the methods and the data.

[Response 0]

We sincerely thank the reviewer's support of our work.

[Comment 1]

The description of the image reconstruction given on pages 3-5 is not transparent enough for the general audience of Nature Communications. I recommend that the authors include a more intuitive description. Leaving the details of the methods to the Methods and Supplementary Materials is OK, but the reader should be able to have a fairly good understanding of how the method works based on the description in the main text. Perhaps the authors can start with an intuitive description of their basic CUP technique and then explain how they add the additional z and polarization dimensions?

[Response 1]

Following the reviewer's suggestions, we have modified the related content in Section 1 of Results. We have added a more explanatory description of SP-CUP's image reconstruction to show how the method works. In particular, we have explained in detail (1) how to recover the (x, y, t) datacube in individual channel (which links to the basic CUP technique), (2) how to compute polarization parameters, and (3) how to extract the depth information. Please see the new content in Lines 114–146 on Pages 4–5 in the revised manuscript.

[Comment 2]

Why is the inverse problem under-determined in this case, and how that affects the image retrieval?

[Response 2]

The under-determined nature of the inverse problem originates from the fact that the number of pixels in the measured data is much fewer than that in the high-dimensional datacube under observation. Under this situation, the sensing matrix does not have a full rank. Simple matrix

inversion cannot faithfully recover the value of each pixel in the datacube. Thus, the inverse problem is under-determined.

However, it has been shown that compressed sensing can accurately solve such an inverse problem via optimization. This salient feature of compressed sensing serves as the foundation of image reconstruction employed in SP-CUP. In a noise-free situation, compressed-sensing algorithms can reconstruct images with excellent image quality. The presence of noise compromises the reconstructed image quality. However, with a reasonable signal-to-noise ratio (SNR), the compressed-sensing algorithm can faithfully recover the scene.

To make our manuscript clearer, we have added more explanation in Section 1 of Results. Please see them in Lines 116–130 on Pages 4–5 in the revised manuscript. We have also realized that the word “under-determined” may be confused with the “incapable of determining the values of reconstructed elements”. Therefore, we have changed the wording to “ill-conditioned”, which is also commonly used in the compressed sensing community [R2].

[Comment 3]

The authors need to include the estimates for the limitations of their imaging technique, within the implementation used here – the x, y, z, t resolutions and ranges, as well as the error of the measurement of the DoLP. What are the major limiting factors? That would give the reader an idea of what type of dynamic processes this technique can be used to image.

[Response 3]

In the original manuscript, the quantification of the spatial (x, y, z) and temporal resolutions have been included, and the results are summarized in Supplementary Table S5 (see Page 37 in the revised manuscript). The ranges of x, y , and t are listed in Summary (see Lines 314–316 on Page 11) and explained (see Lines 841–848 on Page 39) in the revised manuscript.

We have also analyzed the limitations of resolutions in x, y, z , and t in the revised manuscript. In general, the resolutions in x, y , and t are limited by the encoding pixel size, the SNRs in the acquired data, and the overall magnification ratio of the SP-CUP system. The resolution in z is determined by the minimal distinguishable disparity, which is determined by the SNR in the acquired data. We have also added the analysis in the range limitation in x, y, z , and t . The range of (x, y, t) is limited by the sensor size of the internal sensor and the overall magnification ratio of the imaging system. In the design of the SP-CUP system, the range of z is

limited by the depth of field of the camera lens. For detailed analysis of the x , y , z , and t resolutions and ranges, please see Line 816–848 on Pages 37–39 in Supplementary Note 5 in the revised manuscript.

The error of the measurement of the DoLP is mainly induced by the noise performance of the streak camera and by the reconstruction algorithm. Please refer to Response 4 for Reviewer 1. We have experimentally quantified the error of DoLP. The results and analysis are presented in Supplementary Note 6 (see Lines 885–915 on Pages 41–43 in the revised manuscript).

[Comment 4]

In my opinion, including the description of the operation of a streak camera in the Supplementary seems unnecessary. That is a common knowledge.

[Response 4]

We agree with the reviewer that the operating principle of a streak camera is common knowledge to certain communities in physics. However, this knowledge may not be readily available to many other communities that are related to our work, including computational imaging, polarization optics, and three-dimensional imaging acquisition/processing. To make our manuscript a complete and self-referenced piece as well as considering the broad readership of Nature Communication, we decided to keep this section.

[Comment 5]

The author's example involving the polarization evolution of the emission by the ablation plume is interesting, if they are looking at a real effect. First, the ejecta they are showing are not "early-stage plasma," as stated. In fact, the free electrons in the plume are most likely recombined or attached themselves to atoms or oxygen molecules in air within few tens of ps of excitation. It is still not clear why this emission should be polarized. It is commonly believed that this emission (LIBS emission) results from the impact excitation of the ejected atoms and ions by free electrons, followed by unpolarized fluorescence from excited states of those atoms and molecules. The values for the degree of polarization that authors report are not very large. Are they within the measurement error of their method? In other words, are the authors looking at a real effect?

[Response 5]

We thank the reviewer to provide this explanatory comment on our experiment. First, we would like to point out that polarized plasma emission is indeed a real effect. The plasma emission can be divided into two parts from the perspective of spectrum: (1) light from a series of discrete spectral lines and (2) a broadband continuum background. The reviewer is correct that the former, produced by fluorescence, is largely unpolarized. However, it has been shown in numerous literature that the latter is indeed polarized [R3, R4]. This characteristic sparked the development of polarization-resolved laser-induced breakdown spectroscopy (PR-LIBS) to improve the SNR in measured data. More importantly, the emission of the broadband continuum background starts immediately after the laser excitation, while the series of discrete lines appear after tens of nanoseconds [R5]. This characteristic has inspired the development of time-gating-based PR-LIBS to further enhance the signal-to-background ratio [R6]. In our experiment, we imaged the plasma emission from 0 to 3 ns. Therefore, the majority of the light collected by the SP-CUP system is from the broadband continuum background, not from the discrete spectral lines. Thus, the light incident into the SP-CUP system is polarized.

Second, the word “early-stage” refers to the plasma emission, not plasma formation. The same expression has been used in many previous papers. Particularly for our work, we imaged the light from broadband continuum background in plasma emission, not from the discrete spectral lines. Therefore, the term that we used in the manuscript is accurate. However, we realized that in a few places in the original manuscript, “early-stage plasma emission” was not explicated written in certain context. In the revised manuscript, we have used “early-stage plasma emission” consistently throughout the text.

Third, following the reviewer’s suggestion, we have conducted the error quantification of DoLP (see Lines 885–915 on Page 41–43 in Supplementary Note 6 in the revised manuscript). Our results show that the SP-CUP system can offer a highly accurate measurement of DoLP. The decrease in DoLP is 0.25, which is much larger than the value of DoLP reconstruction accuracy (i.e., <0.044). This quantitative comparison supports our claim that the observed decrease in DoLP in the first 3 ns is a real effect.

Fourth, the values of DoLP measured by SP-CUP are supported by the literature. It is known that the polarization state of plasma emission highly depends on the specific experimental setups [R7], including the polarization, pulse width, and pulse energy of the laser beam, the

composition, orientation, and pressure of sample, and the angle of observation. Despite these difficulties, DoLP values of 0.1–0.25 in PR-LIBS using a silicon sample (or a sample whose element has a similar atomic number to that of silicon) can be found in many references, e.g. [R8–R10]. These results are usually based on time-integrated measurement. Thus, the slightly higher DoLP in our measurements results from SP-CUP’s picosecond temporal resolution that excludes the unpolarized emission in discrete spectral lines. In addition, we observed the DoLP’s decrease over time. This trend well agrees with the fact that the polarized light from the broadband continuum background precedes the unpolarized light from the discrete spectral lines.

Finally, possible reasons for polarized broadband plasma emission are discussed in Section 2 of Discussion and are included in many references. These possible reasons include polarization from the incident laser beam, sample reflection of plasma emission, anisotropy in the electron density function, and dynamic polarizability of the core of atoms/ions. According to the literature, the time scales of electron-electron and electron-ion collision can be up to tens of picoseconds and hundreds of nanoseconds [R11], which are within the time window (i.e. 3 ns) of the SP-CUP system. Our results suggest that the polarization in early-stage plasma emission could be determined by the anisotropic electron distribution function in the plasma that decreases homogeneously over time. However, it is important to point out that the exact origin of polarized broadband plasma emission remains elusive. More research is required for detailed studies in each possible mechanisms and their relationship to experimental conditions. The thorough investigation of the origin of polarized plasma emission, however, is certainly beyond the scope of our work. Therefore, we focused on the presentation of experimental results rather than analyzing the underlying mechanisms. Rather, we hope the picosecond temporal resolution and single-shot capability of SP-CUP could contribute to further investigation of this interesting topic.

[Comment 6]

The relevant data shown in Figure 3(c) are not clear. Color coding in two dimensions (DoLP and ψ) may not be a good representation of the data in this case. The plots look predominantly grey. Perhaps the authors can change the color mapping or present the data in some other way to make the features they are talking about stand out better?

[Response 6]

This comment echoes Comment 5 of Reviewer 1. In the revised manuscript, we have re-plotted Fig. 3c and Supplementary Movie 2 by using a jet colormap for DoLP and a grayscale colormap for AoLP. In this way, the drop of DoLP is better visualized.

[Comment 7]

If the measured DoLPs are within the measurement error, what could be a possible physical explanation for this effect? What defines the AoP and why is it changing (or not changing – it is not clear from the data) with time?

[Response 7]

The measured decrease in DoLP is much larger than the measurement error. Multiple possible mechanisms could be used to explain this effect. For details of both points, please refer to our response to your Comment 5.

The AoLP, by name, is defined by the angle of linearly polarized light. For our experiment, the averaged AoLP is 0° (i.e., s-polarized). The averaged AoLP remains constant with $\pm 2.5^\circ$ fluctuation in the entire time window. The value and the time-evolution of the measured mean AoLP can be supported by related literature (e.g., [R12-R14]). According to these papers, the unchanged AoLP could be caused by the Fresnel reflection of the plasma light by the sample or by the characteristic of the plasma itself. For the geometry used in our setup (see Fig. 3a in Main Text), the first potential reason can be ruled out. The second reason is possible because the anisotropic electron distribution function may decrease homogeneously over time, resulting in a decreased DoLP and a maintained AoLP [R15]. However, a precise elucidation of this phenomenon is beyond our expertise and the scope of our paper. We hope that our result will provide experimental data for more theoretical investigation in the future.

We have reflected the above response in Discussion in the revised manuscript. Please see Lines 302–304 on Page 10.

[References]

- [R2] Hansen, P.C., *Discrete inverse problems: insight and algorithms*. 2010: SIAM.
- [R3] Sharma, A. and R. Thareja, *Polarization-resolved measurements of picosecond laser-ablated plumes*. *Journal of Applied Physics*, 2005. **98**(3): p. 033304.

- [R4] Asgill, M., et al., *Investigation of polarization effects for nanosecond laser-induced breakdown spectroscopy*. Spectrochimica Acta Part B: Atomic Spectroscopy, 2010. **65**(12): p. 1033-1040.
- [R5] Le Drogoff, B., et al., *Temporal characterization of femtosecond laser pulses induced plasma for spectrochemical analysis of aluminum alloys*. Spectrochimica Acta Part B: Atomic Spectroscopy, 2001. **56**(6): p. 987-1002.
- [R6] Nejad, M.A. and A.E. Majd, *Temporal Evolution of Polarization Resolved Laser-Induced Breakdown Spectroscopy of Cu*. Plasma Chemistry and Plasma Processing, 2020. **40**(1): p. 325-338.
- [R7] Agnes, N., et al., *The high dependence of polarization resolved laser-induced breakdown spectroscopy on experimental conditions*. Chinese Physics B, 2012. **21**(7): p. 074204.
- [R8] Penczak Jr, J.S., et al., *The mechanism for continuum polarization in laser induced breakdown spectroscopy of Si (111)*. Spectrochimica Acta Part B: Atomic Spectroscopy, 2012. **74**: p. 3-10.
- [R9] Liu, Y., et al., *Observation of near total polarization in the ultrafast laser ablation of Si*. Applied Physics Letters, 2008. **93**(16): p. 161502.
- [R10] Jia, L., et al., *The polarization characteristics of single shot nanosecond laser-induced breakdown spectroscopy of Al*. Chinese Physics B, 2013. **22**(4): p. 044206.
- [R11] Aghababaei Nejad, M., M. Soltanolkotabi, and A. Eslami Majd, *Polarization investigation of laser-induced breakdown plasma emission from Al, Cu, Mo, W, and Pb elements using nongated detector*. Journal of Laser Applications, 2018. **30**(2): p. 022005.
- [R12] Majd, A.E., A. Arabanian, and R. Massudi, *Polarization resolved laser induced breakdown spectroscopy by single shot nanosecond pulsed Nd: YAG laser*. Optics and Lasers in Engineering, 2010. **48**(7-8): p. 750-753.
- [R13] Kim, J. and D.-E. Kim, *Measurement of the degree of polarization of the spectra from laser produced recombining Al plasmas*. Physical Review E, 2002. **66**(1): p. 017401.
- [R14] Wubetu, G., et al., *Time resolved anisotropic emission from an aluminium laser produced plasma*. Physics of Plasmas, 2017. **24**(1): p. 013105.
- [R15] Yoneda, H., et al., *Large anisotropy of the electron distribution function in the high-density plasma produced by an ultrashort-pulse UV laser*. Physical Review E, 1997. **56**(1): p. 988.

REVIEWER 3

[Comment 0]

Introduction and Novelty

The manuscript introduces stereo-polarimetric compressed ultrafast photography (SP-CUP), an imaging system capable of recording high-dimensional events in a single exposure at speeds surpassing other single-shot high-dimensional cameras in the literature. The readout speed of current state-of-the-art sensors has limited single-shot temporal imaging to speeds significantly lower than the speed of light. However, SP-CUP bypasses this issue by implementing streak imaging. The system, however, also avoids inherent problems with streak imaging such as low sequence depth and a low field of view as well as producing high dimensional data by combining it with elements of stereoscopy, and polarimetry, and compressed sensing. The manuscript first describes the optical setup used to achieve these results in detail before moving on to a series of experiments that demonstrate their model's ability to capture multiple photon tags and generalizability to different applications. This method is novel in its advancement of existing CUP techniques by enabling simultaneous recording of polarization in three-dimensional space. It also improves upon current methods of single-shot ultrafast imaging techniques by producing a more favorable sequence depth, spatiotemporal resolution, and scalability in high dimensional imaging. The proposed method is clearly described in the context of alternate options and its novel improvements are outlined in detail.

Major Critiques

This manuscript makes it clear that SP-CUP can enable the collection of multiple photon tags in scattering media, and the recording of non-repeatable, single-shot events such as the polarization dynamics of plasma in laser-induced breakdown. The value that this device brings to the scientific community is evident, however, several aspects of this manuscript should be modified for it to match the caliber of other manuscripts published in Nature Communications.

[Response 0]

Dear Tessa, thank you for pointing out the technical novelty of SP-CUP and for your acknowledgment about the evident value that the SP-CUP system brings to the scientific community.

[Comment 1]

Firstly the paper mentions several times that SP-CUP synergizes compressed sensing and streak imaging with stereoscopy and polarimetry. Not only does this statement become redundant throughout the manuscript, but its implications are not fully explored. How streak imaging, stereoscopy, polarimetry, compressed sensing are integrated into the larger system is made clear, however, why these techniques, in particular, were selected and how they contribute to the advancements that the system has made should be fully fleshed out to fully orient the reader.

[Response 1]

We have deleted the repeated claims about the synergy in SP-CUP's system design in the revised manuscript. We have added more explanation about the underlying rationales to choose specific approaches of stereoscopy and polarimetry for the SP-CUP system from the aspects of functionality, compatibility, and flexibility. We have also associated these rationales with specific steps in image reconstruction to improve the clarification of the description. Please see this information in Lines 114–146 on Pages 4–5 in the revised manuscript.

[Comment 2]

Furthermore, the optical setup figure should be more integrated into the main body of text with perhaps less text description and more references to Figure 1 so that it's easier to visualize what the manuscript is referring to when describing each component.

[Response 2]

Following the reviewer's suggestion, we have shortened the caption of Fig. 1. We have added more details in the description of the system and refer to Fig. 1 as much as possible (see Lines 81–113 on Pages 3–4 in the revised manuscript).

[Comment 3]

The discussion section of the paper introduces more applications where SP-CUP could be implemented. This section is lengthy. Though applications discussed should be mentioned, without quantitative evidence for how SP-CUP performs in these environments, an in-depth write up lacks value to justify its length.

[Response 3]

We have significantly shortened the description of these potential applications in the revised manuscript. In particular, in Discussion, we have largely reduced the length of Section 1 and have deleted Section 3 of the original manuscript. These potential applications are briefly mentioned in Summary (see Lines 327–330 on Page 11 in the revised manuscript).

[Comment 4]

Overall, the manuscript in its current format is not yet suitable for publication without major modifications as suggested above. Should the authors consider addressing those comments, the value of SP-CUP can be made succinct and obvious to the scientific community, and I would gladly review again an updated version of the manuscript.

Tessa Curtis

[Response 4]

We firmly believe that our revision has reflected the values of SP-CUP in a more straightforward means to the scientific community. We sincerely hope that this revised manuscript could satisfy your requirements for publishing our work.

REVIEWERS' COMMENTS:

Reviewer #1 (Remarks to the Author):

The authors have addressed all of my comments. Thank you.

Reviewer #2 (Remarks to the Author):

The authors adequately addressed all of the reviewer's concerns. I recommend publication of the paper in its current form.

Reviewer #3 (Remarks to the Author):

Upon reading the manuscript again I found that the authors addressed all of my concerns as well as the concerns of the other editors. The authors are presenting a highly innovative imaging modality capable of recording stereoscopic and polarization information with picosecond temporal resolution. I believe that Single-shot stereo-polarimetric compressed ultrafast photography: Light-speed observation of high-dimensional optical transients with picosecond resolution is ready for publication in Nature Communications.

Response to Reviewers

We thank the reviewers for their insightful comments, which have helped us improve the quality of our manuscript.

Reviewer 1

Comment 1.1. The authors have addressed all of my comments. Thank you.

[Response]: We thank the reviewer for acknowledging that we have addressed all the comments.

Reviewer 2

Comment 2.1. The authors adequately addressed all of the reviewer's concerns. I recommend publication of the paper in its current form.

[Response]: We thank the reviewer for acknowledging that we have correctly amended the manuscript. We also thank the reviewer for recommending publication of our manuscript.

Reviewer 3

Comment 3.1. Upon reading the manuscript again I found that the authors addressed all of my concerns as well as the concerns of the other editors. The authors are presenting a highly innovative imaging modality capable of recording stereoscopic and polarization information with picosecond temporal resolution. I believe that Single-shot stereo-polarimetric compressed ultrafast photography: Light-speed observation of high-dimensional optical transients with picosecond resolution is ready for publication in Nature Communications.

[Response]: We thank the reviewer for acknowledging that we have addressed all the comments. We also thank the reviewer for acknowledging the innovation in our work and recommending publication of our manuscript.